# Effects of Health Belief, Knowledge, and Attitude toward COVID-19 on Prevention Behavior in Health College Students

**DOI:** 10.3390/ijerph19031898

**Published:** 2022-02-08

**Authors:** Hyeon-Young Kim, Sun-Hwa Shin, Eun-Hye Lee

**Affiliations:** 1Department, College of Nursing, Sahmyook University, Seoul 01795, Korea; hyykimm@syu.ac.kr (H.-Y.K.); shinsh@syu.ac.kr (S.-H.S.); 2VR Healthcare Content Lab, Sahmyook University, Seoul 01795, Korea

**Keywords:** COVID-19, knowledge, attitude, prevention, behavior, health belief model

## Abstract

This study aimed to identify the factors affecting the practice of COVID-19 prevention behaviors among college students as future medical workers. A cross-sectional online survey was conducted in September 2021. A total of 526 health college students were included in this study. A hierarchical regression analysis was performed to examine the effect on the practice of COVID-19 prevention behavior. As a result of the analysis, experiences of education on infectious diseases had significant positive effects on the practice of prevention behavior (β = 0.22, *p* < 0.001). Additionally, a higher COVID-19 health belief had a significant positive effect on the practice of prevention behavior (β = 0.15, *p* = 0.004). Increased smoking and drinking among lifestyle changes after COVID-19 had significant negative effects on the practice of prevention behavior compared with decreased physical activity (β = −0.12, *p* = 0.007). Based on these findings, the study discussed the importance of education on the prevention of infectious diseases among health college students, promotion of health beliefs related to infectious diseases, and formation of healthy lifestyle habits in daily life.

## 1. Introduction

In February 2020, coronavirus disease 2019 (COVID-19), as designated by the World Health Organization (WHO), led to a pandemic. The pandemic caused functional changes in society, leading to the restriction of large group gatherings in facilities and institutions. Accordingly, it had a great impact on education as classes and practical activities were conducted online in non-face-to-face settings [1]. However, college students interact not only in schools but also in other settings; they interact with various types of groups in dormitories and part-time work [2]. In-person classes and clinical practice were offered in some schools during the COVID-19 pandemic without proper analysis and preparation of measures for infection prevention, such as infection awareness education [3]. In particular, health college students were frequently in contact with infectious disease patients, possibly leading to the emergence of new cases. Thus, taking infection prevention and management measures to protect college students is very important.

College students show relatively low interest in health-promoting behaviors compared with adults; however, they are more likely to be accepting of changing their habits and lifestyle. Thus, college years are an important time to establish desirable health habits [4]. Furthermore, in college, students gain autonomy and independence while lacking the experience of autonomy, and their behavior can be greatly affected by social pandemics [5]. Therefore, the lack of adequate knowledge about COVID-19 may have made it difficult for them to make correct judgements, increasing their stress and anxiety about the pandemic [6].

The health belief model is a key predictor of individual health behaviors from various perspectives, such as exercise, diet, smoking, drinking, use of medical institutions, and social roles [7]. This framework presents four components: (1) perceived sensitivity (individual vulnerability to health), (2) perceived severity (individual confidence in the severity of disease), (3) perceived benefits (positive attributes of behavior), and (4) perceived barriers (negative aspects of behavior). More recently, the model has suggested that individual beliefs and cues to actions inform behavior [8,9] (Figure 1). The model has been actively used in health campaigns and programs to predict behaviors in various health-related interests, such as studies on disease conditions, diet, smoking, weight loss, and drinking [10,11]. As such, understanding the psychological states that cause specific behaviors in individuals using the health belief model helps to identify the cause of behavior change and seek methods to induce desirable health behaviors. Thus, it is essential to investigate the relationships between the beliefs, attitudes, and health behaviors of individuals.

Health-related knowledge is an individual’s intellectual ability to maintain daily life; it is an important factor affecting health behaviors. Health-related knowledge can help individuals understand their health and establish appropriate attitudes toward health behaviors. To prevent the spread of infectious diseases such as COVID-19 and minimize the harm caused by it, it is fundamental to follow national recommendations for infectious diseases. Health behavior is affected by the knowledge, attitude, and practice of infectious disease prevention behaviors [12]. In a previous study of health college students on infection control and MERS, higher levels of knowledge and positive attitudes led to greater levels of prevention behaviors [13]. In addition, during the current COVID-19 pandemic, high knowledge about COVID-19 infection has been associated with negative attitudes toward behaviors that may potentially transmit the disease [14].

However, as health college students tend to engage in inadequate levels of infection control compared with their knowledge and risk awareness, factors that directly affect prevention behaviors must be investigated to increase the practice of COVID-19 prevention behaviors among them. Moreover, during pandemics, medical workers are at a high risk of contracting infectious diseases in clinical settings. Therefore, it is necessary to assess the knowledge of future health care workers and health students regarding COVID-19 and their attitudes toward the disease.

This study aimed to identify the factors affecting the practice of COVID-19 prevention behaviors among college students who will become future medical workers. This study can provide basic data for education programs aimed at enabling health college students to protect themselves from COVID-19 infection and prevent its spread.

Focusing exclusively on health college students, this study aims to: (1) assess COVID-19 health beliefs, knowledge, attitudes, and prevention behaviors; (2) assess differences in COVID-19 health beliefs, knowledge, attitudes, and prevention behaviors according to the general characteristics of health college students; (3) investigate the relationship between COVID-19 health beliefs, knowledge, attitudes, and practice of prevention behavior; (4) identify the factors affecting COVID-19 prevention behavior.

## 2. Materials and Methods

### 2.1. Research Design

This was a cross-sectional study with an online observational survey. In this study, participants were selected according to selection and exclusion criteria to examine the correlation between their health beliefs, knowledge, attitudes, and COVID-19 preventive behavior.

### 2.2. Study Participants

The participants in this study were health college students from universities in Korea. A nationwide survey was conducted on 17 regions in Korea, including Seoul and Gyeonggi-do Province. Health college students were those who were studying 21 majors and enrolled in health-related departments in universities, as defined by the Ministry of Education. At the time of the survey, students who dropped out or took a leave of absence were excluded from the study subjects. A questionnaire e-mail was randomly sent to a panel of 3000 university students at Embrain Institute based on the regions, and data were collected from 526 participants who completed the questionnaire.

### 2.3. Research Instruments

#### 2.3.1. COVID-19 Health Belief

COVID-19 health belief was measured using the Health Belief Model Applied to Influenza (HBMAI) developed by Erkin and Ozsoy [9], which was modified and supplemented for COVID-19 by Shin [15]. The tool consisted of 29 items in total, including eight items for perceived sensitivity, four items for perceived severity, six items for perceived benefits, eight items for perceived barriers, and three items for action triggers. Each item was evaluated on a seven-point Likert scale, with a higher total score indicating a greater health belief about infection. In the study by Erkin and Ozsoy [9], the Cronbach’s α of the tool was 0.91. In this study, the Cronbach’s α for the tool was 0.88.

#### 2.3.2. COVID-19 Knowledge

COVID-19 knowledge was evaluated using a 15-item tool developed by Taghrir et al. [16] for medical students based on the diagnosis and treatment protocol of the National Health Commission of China, and modified and supplemented by Kim et al. [17]. The tool consisted of 15 items in total, including five items on public prevention behaviors for COVID-19, two items for infection route, two items for prevention and treatment as a health care professional, four items for etiology and symptoms, and two items for definition and diagnosis. Each item was scored with 0 points for “wrong answer” and “do not know” and 1 point for “correct answer”. The score was calculated out of 100 points, and the total score ranged from 0 to 10 points, with a higher score indicating greater COVID-19 knowledge. The Cronbach’s α of the tool was 0.80 at the time of development and 0.83 in this study.

#### 2.3.3. COVID-19 Attitude

A five-item tool developed by Peng et al. [5] for college students was used to assess COVID-19 attitudes. This tool measured attitudes toward person-to-person transmission, return to school, consumption of wild animals, tolerance for infectious disease emergencies, and the effects of infectious disease control measures. Each item was scored with 0 points for “negative”, 1 point for “neutral”, and 2 points for “positive” according to the response. The total score ranged from 0 to 10 points, with a higher score indicating a more positive attitude toward overcoming the COVID-19 pandemic. The Cronbach’s α was 0.82 in the study by Kim et al. [17] and 0.77 in this study.

#### 2.3.4. Practice of COVID-19 Prevention Behavior

A nine-item tool developed by Taghrir et al. [16] for medical students was used to evaluate the practice of COVID-19 prevention behavior. The tool consisted of nine items in total, including five items on reduction of use of public places, one item on cough prevention behaviors, two items on hand hygiene, and one item on information sharing for prevention. Each item was scored with 0 points for “no” and 1 point for “yes” The total score ranged from 0 to 9 points, with a higher score indicating greater practice of COVID-19 prevention behavior. The Cronbach’s α was 0.72 at the time of tool development and 0.70 in this study.

### 2.4. Data Collection and Ethical Consideration

This study was approved by the Institutional Review Board (IRB) of Sahmyook University in Seoul, and the data were collected through an online questionnaire. The questionnaire was conducted on September 2021. A “research participant explanation” was displayed on the start screen of the questionnaire. The purpose, content, method, voluntary participation, benefits, and risk of participation were explained. In addition, the participants were informed that they could withdraw their consent to participate at any time and that there would be no disadvantages from withdrawal of consent. The students who consented to participate in the questionnaire participated and received an online gift after completing the questionnaire.

### 2.5. Data Analysis

General characteristics, frequency analysis, and descriptive statistics of the key variables are presented. An independent *t*-test was conducted for differences in the study variables according to the general characteristics of the participants, and ANOVA using the Scheffé test was performed for post-hoc analysis. Hierarchical regression analysis was conducted to identify key variables that predicted the moderating effects of the practice of COVID-19 prevention behavior.

## 3. Results

### 3.1. General Characteristics of Participants

The general characteristics of the study participants are listed in Table 1. A total of 429 (81.6%) participants were female, and most of the participants (398 participants; 75.7%) majored in nursing. There were 69 (13.1%), 173 (32.9%), and 284 (54.0%) participants in years 1–2, 3, and 4–5, respectively. In addition, the greatest number of participants had a GPA of 3.5~3.9 (225 participants: 42.8%). There were 219 (41.6%) and 215 (40.9%) participants who had “good” and “normal” subjective health status during the COVID-19 pandemic, respectively. Changes in lifestyle after COVID-19 were as follows: 386 (73.4%) participants experienced decreased physical activity, 81 (15.4%) participants had irregular sleep and dietary habits, 49 (9.3%) participants observed increased eye fatigue, and 10 (1.9%) participants showed increased smoking and drinking behaviors. Furthermore, the majority of the participants had experience in infectious disease prevention education (469 participants; 89.2%), and 46 (8.7%) had been previously infected with COVID-19.

### 3.2. Differences in COVID-19 Health Belief, Knowledge, Attitude, and Practice of Prevention Behavior according to General Characteristics

Differences in COVID-19 health beliefs, knowledge, attitudes, and prevention behaviors according to general characteristics are shown in Table 2. Knowledge was significantly different according to subjective health status during the COVID-19 pandemic (F = 5.49, *p* = 0.004) and experiences of education on infectious diseases (t = 2.45, *p* = 0.014). Post-hoc analysis showed that the group with good subjective health status had higher COVID-19 knowledge than the group with normal subjective health status. Attitude was significantly different according to sex (t = −3.04, *p* = 0.002), changes in lifestyle after COVID-19 (F = 2.75, *p* = 0.042), and experience of education on infectious diseases (t = 2.12, *p* = 0.035). Post-hoc analysis showed no differences in attitudes according to lifestyle changes. Health beliefs were significantly different according to sex (t = −5.44, *p* < 0.001) and grades (F = 4.35, *p* = 0.005). Post-hoc analysis showed that the group with a grade ≥ 4.0 had a higher score of health belief than the group with a grade of 2.9 or less. The practice of prevention behavior was significantly different according to lifestyle changes after COVID-19 (F = 3.04, *p* = 0.029) and experiences of education on infectious diseases (t = 5.28, *p* < 0.001). Post-hoc analysis showed no differences in the practice of COVID-19 prevention behavior according to changes in lifestyle.

### 3.3. Correlation between COVID-19 Health Belief, Knowledge, Attitude, and Practice of Prevention Behavior

The mean and standard deviation of scores for COVID-19 health beliefs, knowledge, attitude, and practice of prevention behavior are shown in Table 3. The mean scores for COVID-19 health belief, knowledge, attitude, and practice of prevention behavior were 4.82 (±0.58), 10.84 (±2.29, out of 15 points), 5.38 (±0.64), and 6.95 (±1.28) points, respectively.

Table 3 shows the correlation between COVID-19 health beliefs, knowledge, attitudes, and practice of prevention behavior. Knowledge was significantly positively correlated with health beliefs (r = 0.10, *p* = 0.026), attitude (r = 0.16, *p* < 0.001), and practice of prevention behavior (r = 0.11, *p* = 0.012). Attitude was significantly positively correlated with health beliefs (r = 0.51, *p* < 0.001) and practice of prevention behavior (r = 0.12, *p* = 0.008). Health beliefs were significantly positively correlated with the practice of prevention behavior (r = 0.17, *p* < 0.001).

### 3.4. Effects of COVID-19 Health Belief, Knowledge, and Attitude on Prevention Behavior

Hierarchical regression analysis was conducted to analyze the effects of health beliefs, knowledge, and attitudes on the practice of prevention behavior, after controlling for the general characteristics of the participants (Table 4). Model 1, which controlled for all general characteristics and included COVID-19-related characteristics and health beliefs, was statistically significant. The explanatory power of the model was 11.0% (R^2^ = 0.110, F = 4.19, *p* < 0.001), and COVID-19 health beliefs had significant effects on the practice of prevention behaviors (β = 0.16, *p* ≤ 0.001). Model 2, which controlled for all general characteristics and COVID-19 health beliefs and included COVID-19 knowledge, had an increased explanatory power of 11.3% (R^2^ = 0.113, F = 4.05, *p* < 0.001). COVID-19 knowledge did not have a significant positive effect on the practice of prevention behavior (β = 0.06, *p* = 0.178). Lastly, Model 3 controlled for all general characteristics, COVID-19 health beliefs, and knowledge, and included COVID-19 attitude. The model had a significant explanatory power of 11.3% (R^2^ = 0.113, F = 3.82, *p* < 0.001). Multicollinearity analysis of the independent variables showed a variance inflation factor (VIF) of 1.02~4.20, satisfying the assumption of independence. The Durbin-Watson value was 1.96, suggesting the independence of the error term. In Model 3, increased smoking and drinking among lifestyle changes after COVID-19 had significant negative effects on the practice of prevention behavior compared with decreased physical activity (β = −0.12, *p* = 0.007). Experiences of education on infectious diseases had significant positive effects on the practice of prevention behavior (β = 0.22, *p* < 0.001). Additionally, a higher COVID-19 health belief had a significant positive effect on the practice of prevention behavior (β = 0.15, *p* = 0.004).

## 4. Discussion

The COVID-19 health beliefs had significant positive effects on prevention behavior. This finding is consistent with that of a previous study in which health beliefs about new infectious diseases had positive effects on hygiene practices for the prevention of infections [15]. This suggests that higher health beliefs about COVID-19 increase the practice of prevention behaviors among health college students. Our data agree with a previous study that showed that health belief increased the intention to get a COVID-19 vaccination and pay for it [18]. Moreover, a previous study reported similar findings that prevention behaviors against COVID-19 infection may be effectively promoted through the health belief model [19]. Another study showed that educational interventions based on the health belief model for COVID-19 are effective in raising awareness and health beliefs among nursing students [20]. Thus, new infectious disease education requires innovative approaches to increase health beliefs.

In this study, practice of COVID-19 prevention behavior in health college students was 6.95 (±1.28) points out of 9 points, which was slightly lower than 7.65 (±1.28) points observed in a previous study conducted approximately a year ago [17]. This suggests possible fatigue from continued prevention behaviors due to the prolonged COVID-19 pandemic. Therefore, it is important to continuously highlight the practice of COVID-19 prevention behaviors for college students majoring in health-related studies who need to pay more attention to infection activities. There were also gender differences in the practice of prevention behavior. Female students showed a more positive attitude than male students, which is consistent with the findings of a previous study on adults [21]. On the other hand, in a study on vaccine administration, opposite findings were observed [22]. This study had a limitation in that the majority of the participant sample (around 82%) were female. Therefore, it is necessary to obtain and analyze more data from male health students in future research. Our results showed that knowledge was positively correlated with attitude, health beliefs, and practice of prevention behavior. This partially supported the findings of a previous cross-sectional study of university students who showed a positive correlation between the practice of COVID-19 prevention behavior and attitude [5]. In our study, the correlation coefficient between COVID-19 health belief and attitude was 0.5, which was significantly higher than the correlation coefficient of 0.34 between hygiene attitude and perceived susceptibility of health belief in nursing students in a previous study [23]. However, although there were correlations between the key variables, the overall regression analysis showed that COVID-19 knowledge and attitude did not affect the practice of prevention behavior. This was in contrast to a previous study that showed the positive effects of knowledge and attitude on the practice of prevention behavior [17]. In another study, educational intervention based on the health belief model increased the self-efficacy of nursing students against COVID-19 [24]. This indicates that health beliefs, rather than knowledge and attitude, about infectious diseases may strengthen the practice of COVID-19 prevention behavior. As health beliefs influence the formation of healthy lifestyles [25], it is important to establish infectious-disease-related health beliefs through infection education in health college students, and active approaches are needed to increase infectious-disease-related health beliefs.

Among participant characteristics, the most influential factor in the practice of COVID-19 prevention behavior was the experience of education on infectious diseases. Previous studies have shown that systemic infection control education at universities can actively promote infection prevention behaviors with correct knowledge and positive attitudes about infectious diseases and may be an effective countermeasure for new infectious diseases [13,26]. In other studies of nursing students, there were significant positive differences in the practice of infection prevention behaviors according to the experience of education on new infectious diseases and intention to participate in prevention education [27,28]. As the intention to participate in new infectious disease prevention education may increase the practice of prevention behaviors, it would be necessary to develop a systematic preventive education program at different levels of health care related to new infectious diseases and provide continued education. In the U.S., the Centers for Disease Control and Prevention continuously provide education to local public health and health care professionals through the Emerging Infections Program [29]. To standardize long-term infectious disease prevention education, infectious disease epidemiology and prevention education programs must be developed and promoted by the government. Efforts should be made to develop education programs for infection prevention and control guidelines for health college students and encourage their proper behaviors. Additionally, education on infectious diseases had the greatest effect on the practice of COVID-19 prevention behavior. Therefore, regular education on infectious diseases is an urgent task for health education among health college students.

Regarding lifestyle changes, the students engaged in increased drinking and smoking as online university life and social distancing became a daily routine; consequently, the practice of COVID-19 prevention behavior reduced. During the prolonged COVID-19 pandemic, college students experienced changes in their lifestyles. Unhealthy behaviors such as drinking, smoking, and reduced physical activity were observed [30,31], which is consistent with our findings. In contrast, although decreased physical activity may be unhealthy behavior, in our study, such a static lifestyle that obeys social distance has reduced possible exposure to infectious diseases, positively affecting the practice of COVID-19 prevention behavior. In a previous study, young adults who were intermittent smokers and unemployed reduced smoking during the COVID-19 pandemic; however, psychological stress increased cigarette consumption [32]. Masks are not worn during smoking, and increased smoking behavior may frequently violate COVID-19 prevention behaviors. Additionally, repeated smoking led to less practice of COVID-19 prevention behaviors. Smokers may smoke in groups and may be exposed to smoke during group dinners. Therefore, infectious diseases must be carefully managed in these groups with reduced sensitivity and awareness of social distancing.

In general, the explanatory power of regression analysis should be 20% or more; however, in our study, the regression model had an explanatory power of 11.3%. This may be attributed to the high health beliefs, knowledge, and attitudes of the health college students in this study, compared with other general groups. In addition, some major variables and general characteristics that may explain infectious disease prevention behaviors in health college students could have been excluded as well. This was a preliminary study on health college students two years after the COVID-19 pandemic, which may have limited the explanatory power; this could be a limitation of this study.

## 5. Conclusions

Experiences of education on infectious diseases had the greatest effects on the practice of COVID-19 prevention behavior in health college students, followed by health beliefs. On the other hand, increased smoking and drinking habits had negative effects on the practice of COVID-19 prevention behaviors. Altogether, the findings of this study emphasize the urgent need to develop and implement active infectious disease prevention education and promote infectious disease health beliefs in health and medical students. Additionally, it is important to help health college students form healthy daily lifestyles during college years when their healthy lifestyles are formed.

## Figures and Tables

**Figure 1 ijerph-19-01898-f001:**
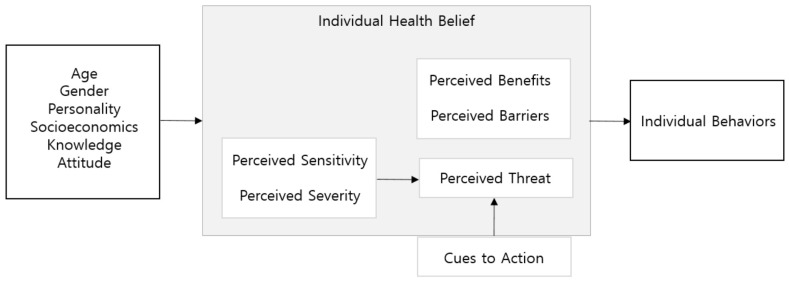
The Health Belief Model.

**Table 1 ijerph-19-01898-t001:** General Characteristics of Participants (*n* = 526).

Characteristics	Categories	*n* (%)
Gender	Male	97 (18.4)
	Female	429 (81.6)
Major	Nursing	398 (75.7)
	Non-nursing	128 (24.3)
Academic year	1~2	69 (13.1)
	3	173 (32.9)
	4~5	284 (54.0)
Grade point average	<2.9	37 (7.0)
	3.0~3.4	123 (23.4)
	3.5~3.9	225 (42.8)
	≥4.0	141 (26.8)
Subjective health status during the COVID-19 pandemic	Bad	92 (17.5)
	Normal	215 (40.9)
	Good	219 (41.6)
Changes in lifestyle after COVID-19	Decreased physical activity	386 (73.4)
	Increased smoking and drinking behavior	10 (1.9)
	Irregular sleep and dietary habits	81 (15.4)
	Increased eye fatigue	49 (9.3)
Experience of education on infectious diseases	Yes	469 (89.2)
	No	57 (10.8)
Experience of previous infection with COVID-19	Yes	46 (8.7)
	No	480 (91.3)

**Table 2 ijerph-19-01898-t002:** Differences in COVID-19 Health Belief, Knowledge, Attitude, and Practice of Prevention Behavior according to General Characteristics (*n* = 526).

Characteristics	Categories	COVID-19 Knowledge		COVID-19 Attitude		COVID-19 HB		COVID-19 PPB	
		M ± SD	t/F(*p*)	M ± SD	t/F(*p*)	M ± SD	t/F(*p*)	M ± SD	t/F(*p*)
Gender	Male	10.69 ± 2.79	−0.71 (0.477)	5.20 ± 0.76	−3.04 (0.002)	4.53 ± 0.60	−5.44 (<0.001)	6.88 ± 1.60	−0.60 (0.548)
	Female	10.87 ± 2.17		5.42 ± 0.61		4.88 ± 0.56		6.96 ± 1.20	
Major	Nursing	10.74 ± 2.38	−1.75 (0.080)	5.38 ± 0.65	0.21 (0.832)	4.81 ± 0.57	−0.18 (0.861)	6.99 ± 1.22	1.29 (0.199)
	Non-nursing	11.15 ± 1.96		5.37 ± 0.64		4.82 ± 0.62		6.82 ± 1.43	
Academic year	1~2	10.71 ± 2.36	2.29 (0.103)	5.44 ± 0.80	0.41 (0.667)	4.92 ± 0.60	1.55 (0.213)	6.68 ± 1.32	1.87 (0.155)
	3	11.14 ± 2.02		5.35 ± 0.65		4.78 ± 0.57		7.03 ± 1.15	
	4~5	10.69 ± 2.42		5.38 ± 0.60		4.81 ± 0.59		6.96 ± 1.34	
Grade point average	<2.9 ^a^	10.11 ± 3.41	2.02 (0.111)	5.21 ± 0.77	1.68 (0.171)	4.58 ± 0.68	4.35 (0.005) a < d	6.70 ± 1.61	1.35 (0.256)
	3.0~3.4 ^b^	10.69 ± 2.44		5.40 ± 0.62		4.80 ± 0.61		6.80 ± 1.28	
	3.5~3.9 ^c^	10.89 ± 2.25		5.35 ± 0.69		4.79 ± 0.54		7.03 ± 1.23	
	≥4.0 ^d^	11.09 ± 1.78		5.45 ± 0.55		4.94 ± 0.57		7.01 ± 1.24	
Subjective health status after COVID-19	Bad ^a^	10.80 ± 1.91	5.49 (0.004) b < c	5.50 ± 0.57	2.10 (0.123)	4.92 ± 0.60	1.97 (0.140)	6.79 ± 1.17	0.96 (0.384)
	Normal ^b^	10.48 ± 2.61		5.34 ± 0.66		4.82 ± 0.52		7.01 ± 1.23	
	Good ^c^	11.21 ± 2.04		5.36 ± 0.65		4.77 ± 0.63		6.95 ± 1.37	
Changes in lifestyle after COVID-19	Decreased physical activity	10.85 ± 2.28	2.29 (0.078)	5.42 ± 0.63	2.75 (0.042)	4.82 ± 0.55	0.47 (0.704)	7.01 ± 1.20	3.04 (0.029)
	Increased smoking and drinking behavior	9.20 ± 2.49		5.55 ± 0.53		4.60 ± 0.39		5.90 ± 2.23	
	Irregular sleep and dietary habits	10.74 ± 2.39		5.24 ± 0.68		4.82 ± 0.68		6.79 ± 1.41	
	Increased eye fatigue	11.24 ± 2.11		5.23 ± 0.67		4.82 ± 0.66		6.90 ± 1.34	
Education on infectious diseases	Yes	10.93 ± 2.27	2.45 (0.014)	5.40 ± 0.64	2.12 (0.035)	4.82 ± 0.58	0.78 (0.437)	7.05 ± 1.15	5.28 (<0.001)
	No	10.14 ± 2.36		5.21 ± 0.65		4.76 ± 0.57		6.12 ± 1.89	
Experience of infection with COVID-19	Yes	11.33 ± 1.96	−1.51 (0.132)	5.37 ± 0.69	0.05 (0.960)	4.76 ± 0.52	0.72 (0.470)	6.98 ± 1.11	−0.18 (0.861)
	No	10.79 ± 2.32		5.38 ± 0.64		4.82 ± 0.59		6.94 ± 1.30	

COVID-19 HB = COVID-19 health beliefs; COVID-19 PPB = COVID-19 practice of prevention behavior.

**Table 3 ijerph-19-01898-t003:** Correlation between COVID-19 Health Belief, Knowledge, Attitude, and Practice of Prevention Behavior (*n* = 526).

Variables	COVID-19 Knowledge	COVID-19 Attitude	COVID-19 PPB	Mean ± SD	Skewness	Kurtosis
	r (*p*)					
COVID-19 HB	0.10 (0.026)	0.51 (<0.001)	0.17 (<0.001)	4.82 ± 0.58	0.01	0.48
COVID-19 knowledge		0.16 (<0.001)	0.11 (0.012)	10.84 ± 2.29	−1.95	5.90
COVID-19 attitude			0.12 (0.008)	5.38 ± 0.64	−0.55	0.66
COVID-19 PPB				6.95 ± 1.28	−1.67	3.77

COVID-19 HB = COVID-19 health beliefs; COVID-19 PPB = COVID-19 practice of prevention behavior.

**Table 4 ijerph-19-01898-t004:** Effects of COVID-19 Health Belief, Knowledge, and Attitude on Prevention Behavior (*n* = 526).

Variables (Reference)	Model 1			Model 2			Model 3			
	b	β	*p*	b	β	*p*	b	β	*p*	VIF
Changes in lifestyle after COVID-19 (Decreased physical activity)										
Increased smoking and drinking behavior	−1.11	−0.12	0.006	−1.07	−0.11	0.008	−1.08	−0.12	0.007	1.07
Irregular sleep and dietary habits	−0.16	−0.05	0.286	−0.16	−0.04	0.301	−0.15	−0.04	0.332	1.07
Increased eye fatigue	−0.04	−0.01	0.849	−0.05	−0.01	0.801	−0.04	−0.01	0.842	1.07
Education on infectious diseases (No)	0.93	0.23	<0.001	0.90	0.22	<0.001	0.89	0.22	<0.001	1.10
Experience of infection with COVID-19 (No)	0.06	0.01	0.773	0.03	0.01	0.858	0.03	0.01	0.858	1.04
COVID-19 health beliefs	0.36	0.16	<0.001	0.35	0.16	<0.001	0.32	0.15	0.004	1.45
COVID-19 knowledge				0.03	0.06	0.178	0.03	0.06	0.204	1.11
COVID-19 attitude							0.05	0.02	0.618	1.42
R^2^ (ΔR^2^)	0.110 (0.024)			0.113 (0.003)			0.113 (<0.001)			
F (*p*)	4.19 (<0.001)			4.05 (<0.001)			3.82 (<0.001)			

Note. Adjusted for Gender, Major, Academic year, Grades and Subjective health status.

## Data Availability

The data presented in this study are available on request from the corresponding author.

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
