# Peer review of "Effects of Health Belief, Knowledge, and Attitude toward COVID-19 on Prevention Behavior in Health College Students"

_ijerph, 2022, doi:10.3390/ijerph19031898_

Round 1

Reviewer 1 Report

The article is of undoubted interest for the scientific community in the field of health. It investigates a problem of worldwide claim today and presents some results that affect the value of beliefs for human behavior, in this case of health prevention. These results connect with the importance of beliefs, in general, for people’s behaviors. It seems to us like a relevant contribution.

The work is correctly structured and developed with sufficient rigour. To point out some improvements that could be introduced in the text, we indicate the following. The research design, defined as a “cross-sectional study”, could be explained in greater detail. Also, the sample design could be better defined, delimiting the distribution of universities and regions involved in the study.

In general, we believe that the article meets sufficient conditions to be published.

Author Response

Dear Editor and Reviewers,

We wish to thank you for your thoughtful comments and valuable feedback on the manuscript originally titled, “Effects of Health Belief, Knowledge, and Attitude Toward COVID-19 on Prevention Behavior in Health College Students”

We have modified the manuscript according to your suggestions, rewriting and rephrasing sections to improve clarity, adding further information, and explaining in detail the points that were previously vague. For your convenience, we have set the revisions in the manuscript in red. We believe that the revised version of this paper will be of interest to the readership of the International Journal of Environmental Research and Public Health.

-----------------------------------------------------------------------------------------------------

Reviewer 1’s comment

Comments and Suggestions for Authors

Point 1. The article is of undoubted interest for the scientific community in the field of health. It investigates a problem of worldwide claim today and presents some results that affect the value of beliefs for human behavior, in this case of health prevention. These results connect with the importance of beliefs, in general, for people’s behaviors. It seems to us like a relevant contribution.

  • Response: Thank you for your meaningful comments.

Point 2. The work is correctly structured and developed with sufficient rigour. To point out some improvements that could be introduced in the text, we indicate the following. The research design, defined as a “cross-sectional study”, could be explained in greater detail.

  • Response: Thank you for your valuable comments. We have explained in greater detail as per your suggestion.

On page 2:

  • Lines 90-93: This was a cross-sectional study with an online observational survey. In this study, participants were selected according to selection and exclusion criteria to examine the correlation between participants' health beliefs, knowledge, attitudes, and COVID-19 preventive behavior.

Point 3.  Also, the sample design could be better defined, delimiting the distribution of universities and regions involved in the study.

  • Response: We have rephrased the sentence as per your comments.

On page 3:

  • Lines 96-102: The participants in this study were health college students from universities in Korea. A nationwide survey was conducted on 17 regions in Korea, including Seoul and Gyeonggi-do Province. Health college students were those who were studying 21 majors and enrolled in health-related departments in universities, as defined by the Ministry of Education. At the time of the survey, students who dropped out or took a leave of absence were exclude from the study subjects. A questionnaire e-mail was randomly sent to a panel of 3,000 university students at Embrain Institute based on the regions, and data were collected from 526 participants who completed the questionnaire.

Point 4.  In general, we believe that the article meets sufficient conditions to be published.

  • Response: Thank you very much for your prompt and positive review.

Reviewer 2 Report

The study tested how health belief, knowledge, and attitude towards COVID_19 correlated with the prevention behavior and found that participants education on infectious diseases and health believes had a relationship with the prevention behavior practices.

P1. Paragraph 3 – Health belief model needs to be unpacked here: what specific dimensions/beliefs that are included in the health belief model. Is it the same model as Erkin and Ozsoy? Is this model connected with the survey tool by Erkin and Ozsoy? How? What needs to be unpacked is what does “perceived sensitivity” (of what?), “perceived severity” (of what?), “perceived benefit” (of what?), “perceived disability “(of what?) and “action triggers” (which are?) refers to? And how these 5 dimensions are part of the health belief model. A graph or visualization might be helpful.

P4. 3.1 Participants: “the greatest number of participants had a GPA of less than 3.5~4 (225%) is not true or at least conflicts with the data in the table. Should it be something like majority of the participants had a GPA 3.5 or higher (366 total)?

P8. Line 3 from the top – when talking about gender differences it would be important to note the limitation of this study which cannot be helped i.e. that the majority of the participants sample (around 82%) in this study are female. So the findings speak to better understanding of this population and more data from the male health students need to be obtained in the future.

Related to that where there any noticeable differences in the patterns between the genders in this study? Did the 85 male respondents patterned differently?

Author Response

Dear Editor and Reviewers,

We wish to thank you for your thoughtful comments and valuable feedback on the manuscript originally titled, “Effects of Health Belief, Knowledge, and Attitude Toward COVID-19 on Prevention Behavior in Health College Students”

We have modified the manuscript according to your suggestions, rewriting and rephrasing sections to improve clarity, adding further information, and explaining in detail the points that were previously vague. For your convenience, we have set the revisions in the manuscript in red. We believe that the revised version of this paper will be of interest to the readership of the International Journal of Environmental Research and Public Health.

-----------------------------------------------------------------------------------------------------

Reviewer 2’s comment

Comments and Suggestions for Authors

Point 1. Paragraph 3 – Health belief model needs to be unpacked here: what specific dimensions/beliefs that are included in the health belief model. Is it the same model as Erkin and Ozsoy? Is this model connected with the survey tool by Erkin and Ozsoy? How? What needs to be unpacked is what does “perceived sensitivity” (of what?), “perceived severity” (of what?), “perceived benefit” (of what?), “perceived disability “(of what?) and “action triggers” (which are?) refers to? And how these 5 dimensions are part of the health belief model. A graph or visualization might be helpful.

  • Response: Thank you for your meaningful comments. In paragraph 3, we added a description of the concept of health belief model configuration and inserted figure1 for visualization.

On page 1~2:

  • Lines 44-56: The health belief model is a key predictor of individual health behaviors from various perspectives, such as exercise, diet, smoking, drinking, use of medical institutions, and social roles (7). This framework presents four components: 1) perceived sensitivity (individual vulnerability to health), 2) perceived severity (individual confidence in the severity of disease), 3) perceived benefits (positive attributes of behavior), and 4) perceived barriers (negative aspects of behavior). More recently, the model suggests that individual beliefs and cues to actions inform behavior (8, 9) (Figure 1). The model has been actively used in health campaigns and programs to predict behaviors in various health-related interests, such as studies on disease conditions, diet, smoking, weight loss, and drinking (10, 11). As such, understanding the psychological states that cause specific behaviors in individuals using the health belief model helps to identify the cause of behavior change and seek methods to induce desirable health behaviors. Thus, it is essential to investigate the relationships between the beliefs, attitudes, and health behaviors of individuals.
  • Figure 1

Point 2. P4. 3.1 Participants: “the greatest number of participants had a GPA of less than 3.5~4 (225%) is not true or at least conflicts with the data in the table. Should it be something like majority of the participants had a GPA 3.5 or higher (366 total)?

  • Response: Thank you for your valuable comments. GPA grade distribution surveyed in the text is true, and 366 people (75.3%) were found to be GPA 3.5 or higher. We revised the contents described in the text to match the results of Table 1.

On page 4:

  • Lines 166-167: In addition, the greatest number of participants had a GPA of 3.5∼9 (225 participants: 42.8%).

Point 3. P8. Line 3 from the top – when talking about gender differences it would be important to note the limitation of this study which cannot be helped i.e. that the majority of the participants sample (around 82%) in this study are female. So the findings speak to better understanding of this population and more data from the male health students need to be obtained in the future.

  • Response: Thank you for the suggestion. We have added this point in the discussion section as per your suggestion.

On page 8:

  • Lines 256-262: There were also gender differences in the practice of prevention behavior. Female students showed a more positive attitude than male students, which is consistent with the findings of a previous study on adults (21). On the other hand, in a study on vaccine administration, opposite findings were observed (22). This study has a limitation in that the majority of the participant sample (around 82%) are female. Therefore, it is necessary to be obtained and analyzed more data from the male health students in the future research.

Point 4.  Related to that where there any noticeable differences in the patterns between the genders in this study? Did the 85 male respondents patterned differently?

  • Response: Thank you for considerate review. The purpose of this study is not to observe gender differences and moderating effects. We will consider future research to ensure sufficient information on male health college students and investigate the main characteristics of male students.
